# On-Premises LLM Deployment Demands a Middle Path: Preserving Privacy Without Sacrificing Model Confidentiality

**Hanbo Huang[1], Yihan Li, Bowen Jiang, Lin Liu, Ruoyu Sun[2], Zhuotao Liu[3], Shiyu Liang[1]**
[1]Shanghai Jiao Tong University,
[2]Chinese University of Hong Kong (Shenzhen), [3]Tsinghua University
{hhuang417,lsy18602808513}@sjtu.edu.cn

## Abstract

Current LLM customization typically relies on two deployment strategies: closed-source APIs, which require users to upload private data to external servers, and open-weight models, which allow local fine-tuning but pose misuse risks. In this paper, we argue that (1) deploying closed-source LLMs within user-controlled infrastructure (*on-premises deployment*) enhances data privacy and mitigates misuse risks, and (2) a well-designed on-premises deployment must ensure model confidentiality—by preventing model theft—and offer privacy-preserving customization. Prior research on small models has explored securing only the output layer within hardware-secured devices to balance confidentiality and customization efficiency. However, we show that this approach is insufficient for defending large-scale LLMs against distillation attacks. We therefore introduce a semi-open deployment framework that secures only a few, carefully chosen layers, achieving distillation resistance comparable to fully secured models while preserving fine-tuning flexibility. Through extensive experiments, we show that securing bottom layers significantly reduces functional extraction risks. Our findings demonstrate that privacy and confidentiality can coexist, paving the way for secure on-premises AI deployment that balances usability and protection.

## 1 Introduction

Vendors of Large Language Models (LLMs) have introduced advanced models with remarkable capabilities to address diverse user needs (Minaee et al., 2024). To enable customization and drive industry progress, vendors typically adopt two approaches. Closed-source vendors, like OpenAI, provide fine-tuning APIs that allow users to upload data to customize proprietary models such as GPT-4. In contrast, companies like Meta offer open-weight models, such as Llama3 (Dubey et al., 2024), which users can adapt within their own infrastructure, ensuring greater flexibility and control.

However, both approaches present significant limitations for privacy-sensitive users, such as financial institutions, healthcare organizations, and government agencies, which prioritize data security. Fine-tuning APIs from closed-source vendors offer encryption for sensitive data and comply with strict privacy regulations (Pang et al., 2024). However, their security fundamentally depends on vendor trust, raising ethical concerns and exposing users to potential data breaches (Sun et al., 2023).

Alternatively, fine-tuning open-weight models within private infrastructure—commonly referred to as *on-premises deployment*, where all data and model adaptation occur locally—ensures data protection and customization (Nevo et al., 2024). However, full disclosure of model architectures and weights heightens the risk of misuse by malicious actors, enabling misinformation generation, automated cyberattacks, and security bypasses (Hendrycks et al., 2023). These risks discourage vendors from releasing SOTA models as open-weight, as uncontrolled distribution could lead to widespread harm. Beyond security concerns, maintaining high-quality open-weight models imposes substantial computational and financial costs (Wolfe et al., 2024), further disincentivizing full disclosure.

In this paper, we argue that **deploying closed-source SOTA models on-premises with restricted access to authorized users offers a middle path to mitigate privacy leakage and model misuse**

**risks.** However, prior research has shown that hardware-based extraction attacks can recover model parameters and architectures directly from CPUs and memory (Hu et al., 2020) within local infrastructures, enabling unauthorized users to steal proprietary model information from the deployment server. This underscores the need for stronger security measures. Therefore, we further argue that **a well-designed on-premises deployment strategy must fully prevent model theft while preserving data privacy, ensuring no reliance on untrusted servers.**

Our work provides an affirmative step toward addressing the fundamental challenge of balancing security and customization in on-premises LLM deployment. Existing approaches have sought to protect proprietary models using Trusted Execution Environments (TEEs) (Narra et al., 2019), but fully enclosing models within TEEs incurs prohibitive computational costs, limiting their practicality (Nayan et al., 2024). Prior research has attempted to mitigate this trade-off by securing only critical layers, such as the output layer, while leaving the remaining layers exposed for

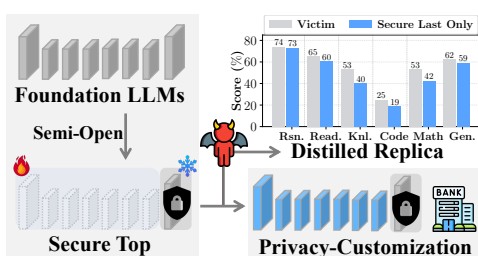

Figure 1: Semi-open Deployment.

fine-tuning (Zhang et al., 2024b; Mo et al., 2020). However, our findings align with prior work showing that securing only the output layer is insufficient against distillation attacks (Zanella-Beguelin et al., 2021). Extending these attacks to Llama2-70B, we confirm that this vulnerability persists at scale, enabling near-complete functionality distillation in six domains, as shown in Figure 1. These vulnerabilities raise skepticism about *whether model confidentiality and customization can truly coexist in on-premises deployment*, highlighting the need for a security paradigm beyond final-layer protection. Without a viable solution, vendors risk intellectual property theft, while users must compromise data privacy when relying on external servers.

Our work suggests that this dilemma can be resolved. We propose a theoretically inspired semi-open deployment strategy that secures only a few, carefully chosen layers, balancing security and customization while mitigating distillation risks. This enables secure on-premises AI adoption in privacy-sensitive industries such as healthcare, finance, and government. More broadly, it highlights the need for hybrid security solutions that balance usability and protection, lowering barriers to responsible AI deployment.

By demonstrating that confidentiality and customization can coexist, our work shifts the narrative from skepticism to opportunity, paving the way for AI-driven innovation in regulated sectors. Future research should explore optimal layer selection, adaptive security mechanisms, and distillation-resistant architectures to further enhance model confidentiality. We hope this work inspires the AI community to develop scalable solutions that ensure AI remains both secure and widely accessible. We summarize our contribution as follows.

- **Semi-open on-premises LLM deployment.** We propose a semi-open deployment framework that secures a single, carefully chosen layer, achieving distillation resistance comparable to full encryption while preserving customization flexibility comparable to full parameter fine-tuning.
- **Layer-wise security transitions.** We provide the first theoretical result identifying a transition layer that resists distillation attacks, revealing fundamental differences in security across layers.
- **Comprehensive empirical evaluations.** We have conducted extensive empirical evaluations across five models (Llama2 (70B,7B), Mistral-7B, Phi-2 (2.7B), and Phi-1.5 (1.3B) ) under three distillation strategies with a 51k attack set, comparing its performance with three baselines.

## 2  RELATED WORKS

**Data Privacy Risks.** Using LLM services for customization introduces significant data privacy risks, as user data is exposed during transmission, storage, and processing (Li et al., 2024b). To address this, prior research has explored Homomorphic Encryption (Lee et al., 2022), which allows computations on encrypted data, and Differential Privacy (Wei et al., 2021), which injects noise to prevent sensitive data memorization. However, these methods suffer from high computational overhead (Zhou et al., 2024) and limited protection coverage (Wei et al., 2020). A more effective approach is *on-premises deployment*, which ensures data remains under user control (Nevo et al., 2024). While this mitigates

privacy concerns, it shifts the risk to LLM vendors, who lose control over model usage and face heightened risks of theft and misuse (Hendrycks et al., 2023).

**Model Theft Risks** Deploying LLMs on user-controlled infrastructure increases exposure to model extraction threats (Atli et al., 2020). Attackers can recover model parameters through hardware-based techniques, including side-channel attacks targeting GPUs (Nayan et al., 2024) and bus monitoring attacks (Hu et al., 2020). In addition to direct parameter extraction, *distillation attacks* allow adversaries to replicate model functionality using only black-box access, a process known as functional distillation (Nevo et al., 2024; Xu et al., 2024; Ezzeddine et al., 2024). While distillation attacks have been extensively studied in smaller models, such as CNNs (Orekondy et al., 2018), BERT (Sanh et al., 2020; Zanella-Beguelin et al., 2021), and ReLU-based models (Canales-Martínez et al., 2024; Jagielski et al., 2020), their effectiveness against large-scale LLMs remains an open question. Our work extends these attacks to Llama2-70B and demonstrates that securing only the output layer remains insufficient to prevent near-complete functionality replication.

**Traditional Approaches.** Balancing data privacy and model security in on-premises deployment requires novel solutions. One direction is *privacy-preserving federated deployment*, where a model is split between user-controlled infrastructure and vendor-managed servers (Shen et al., 2023; Huang et al., 2024). Users fine-tune local layers while the vendor controls deeper layers, with privacy-preserving techniques like differential privacy (Zhou et al., 2024) protecting sensitive data. However, industries with strict data regulations, such as banking, healthcare, and government, often require full on-premises deployment (Schillaci, 2024; Guerra-Manzanares et al., 2023), ensuring models remain within their infrastructure while allowing customization. To secure models in on-premises settings, research has explored *hardware-based protections* such as Arm TrustZone (Pinto & Santos, 2019) and secure execution environments (Zhang et al., 2024a; Li et al., 2024a). These approaches isolate computations to prevent unauthorized extraction but impose high resource demands and restrict customization (Mo et al., 2020). A more flexible alternative is *layer-wise security*, where only critical layers are protected while others remain exposed (Lin et al., 2024; Chen et al., 2024; Zhang et al., 2024b). Studies suggest different strategies, including securing shallow layers (Elgamal & Nahrstedt, 2020), intermediate layers (Shen et al., 2022), or the output layer (Huang et al., 2024). However, most research has focused on smaller models, leaving the effectiveness of different security placements in large language models unclear.

# 3 PRELIMINARIES

## 3.1 SECURITY THREAT: MODEL DISTILLATION

**Adversary's Objective.** The adversary aims to replicate the functionality of a semi-open victim LLM, partially secured in a protected environment, by building a replacement model. The agreement between the victim and replicated models is measured through fidelity on specified testing datasets.

**Adversary's Knowledge.** It is assumed that the adversary knows the architectures of both secured and unsecured modules, as prior work (Gou et al., 2021; Boix-Adsera, 2024) has shown that using the same architecture as the secured module significantly improves the effectiveness of distillation attacks. However, the adversary knows only the parameters of the unsecured module, while those of the secured module remain unknown due to their concealment within hardware-secured environments. Additionally, given that LLMs are typically trained on proprietary data derived from publicly available sources (Dubey et al., 2024), we assume that the adversary does not have direct access to the exact training data of the victim model, but is aware of its approximate distribution.

**Adversary's Capability.** The adversary is capable of querying the semi-open model, obtaining both the semantic output produced by the complete model and the representation vector generated by the secured module. Utilizing this information, the adversary constructs a distillation attack dataset denoted as $\mathcal{D}$. Since the adversary knows the architecture of the secured module, the adversary next replaces the secured module with a randomly initialized module of

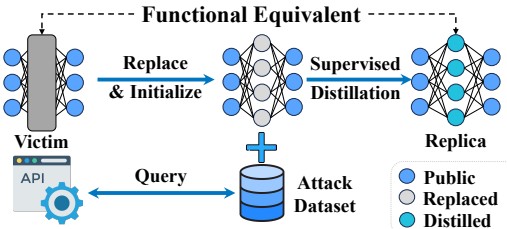

Figure 2: Workflow of model distillation attack

the same architecture. Using the constructed set $\mathcal{D}$, the adversary employs three distinct supervised distillation strategies to replicate the functionality of the secured module: (1) **FT-all:** Fine-tunes both the replacement and unsecured modules using output of the entire model as training labels. (2) **FT-closed:** Fine-tunes only the replacement model using output of the entire model, keeping the unsecured module fixed. (3) **SEM** (Tamber et al., 2024): Fine-tunes the replacement model using outputs from the secured module without involving the unsecured component.

## 3.2 Problem Formulation

In this paper, we analyze the performance of a large language model under a defined distribution $\mathbb{P}_{\mathbf{X} \times Y}$, describing the relationship between input matrix $\mathbf{X}$ and label $Y$. We assume the victim LLM $f(\mathbf{X}; \boldsymbol{\theta})$ performs well on this distribution, and the attack set $\mathcal{D}$ comprises samples drawn from $\mathbb{P}_{\mathbf{X} \times Y}$. To evaluate agreement between the distilled LLM and ground-truth labels, we use a scoring function $s : \mathcal{Y} \times \mathcal{Y} \to \mathbb{R}^+$. Secured layers are indexed by $I \subseteq [L] = \{1, \ldots, L\}$. Let $\boldsymbol{\theta}_{\text{dist}}(I, \mathcal{D})$ represent the parameter vector of the distilled replica of a victim model, where layers indexed by $I$ are secured, and adversaries utilize the attack set $\mathcal{D}$ to replicate its functionality. For each secured set $I$, we define the "**Distillation Ratio**" $R(I)$, which quantifies how well the distilled model $\boldsymbol{\theta}_{\text{dist}}(I, \mathcal{D})$ replicates the behavior of $f(\mathbf{X}; \boldsymbol{\theta})$, expressed as

$$R(I) = \frac{\mathbb{E}[s(f(\mathbf{X}; \boldsymbol{\theta}_{\text{dist}}(I, \mathcal{D})), Y)]}{\mathbb{E}[s(f(\mathbf{X}; \boldsymbol{\theta}), Y)]}. \tag{1}$$

Here, $\mathbb{E}$ in the numerator reflects the expectation computed over random samples $(\mathbf{X}, Y)$ drawn from $\mathbb{P}_{\mathbf{X} \times Y}$, the random attack set $\mathcal{D}$, and the random initialization of parameters within the secured layers during fine-tuning. Conversely, the term $\mathbb{E}$ in the denominator solely considers the expectation over random samples. With this definition, $R([L])$ represents the distillation ratio when the entire model is secured, reflecting the highest level of security. This leads to the question:

> What is the smallest secured set $I$ such that $R(I)$ closely approximates $R([L])$?

This question aims to identify the minimal secured set $I$ such that securing the layers indexed by $I$ achieves a level of security comparable to securing the entire model.

## 4 Methodology

In this section, we investigate the impact of securing specific layers on security and customization against distillation attacks. We begin with an experiment with two semi-open deployments of Llama2-70B: one securing the bottom two decoder layers (Bottom2-Secured) and the other securing the top two decoder layers (Top2-Secured). As shown in Figure 3, both deployments achieve similar customization performance in six downstream tasks. However, securing the bottom layers provides significantly stronger security. Additionally, comparing Bottom2-Secured to fully-secured deploy-

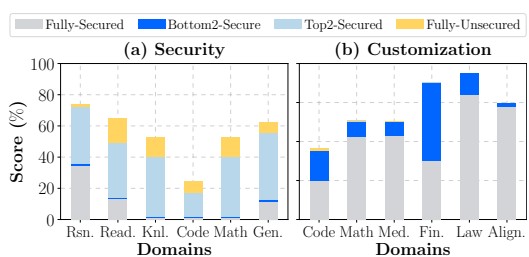

Figure 3: Security and adaptability comparison in Llama2-70B. Lower scores indicate better security in Fig.(a) and weaker adaptability in Fig.(b).

ment reveals comparable security with improved customizability. This suggests that securing a certain number of bottom layers can effectively balance strong security against distillation attacks and high customization performance.

### 4.1 Security Transition in Deep Transformers

**Model Overview.** In this subsection, we consider a deep transformer $f$ with $L$ layers, expressed as $f(\mathbf{X}; \boldsymbol{\theta}) = \varphi_L \circ \cdots \circ \varphi_1(\mathbf{X})$. The input feature matrix $\mathbf{X} \in \mathbb{R}^{n \times d}$ consists of $n$ rows, each representing a $d$-dimensional token vector. Each layer $\varphi_i$ is a transformer that incorporates a normalized residual self-attention mechanism, defined as:

$$\varphi_i(\mathbf{X}; K_i, Q_i) = \mathbf{X} + \text{softmax}\left(\frac{\mathbf{X}Q_i(\mathbf{X}K_i)^\top}{\sqrt{d_Q}\|\mathbf{X}\|^2}\right)\mathbf{X}.$$

Here, $Q_i \in \mathbb{R}^{d \times d_Q}$ and $K_i \in \mathbb{R}^{d \times d_Q}$ are projection matrices for the query and key components, respectively. The terms $\sqrt{d_Q}$ and $\|\mathbf{X}\|$ serve as normalization factors, ensuring stable computations within the attention mechanism. We consider the semi-open deployment of securing the $\alpha L$-th layer with $\alpha \in [0, 1]$ and $\alpha L \in \mathbb{N}$ while keeping other layers unsecured. After the distillation attack, we assume the parameters of the distilled model in the unsecured layers are identical to the victim model, while those in the secured layer deviate. Let $\hat{K}_{\alpha L}$ and $\hat{Q}_{\alpha L}$ denote the distilled weight matrix of the proprietary layer, i.e., $\boldsymbol{\theta}_{\text{dist}}(\{\alpha L\}) = \{(K_1, Q_1), ..., (\hat{K}_{\alpha L}, \hat{Q}_{\alpha L}), ..., (K_L, Q_L)\}$. Let $\hat{\varphi}_{\alpha L}$ denote the function of the distilled proprietary layer, i.e., the $\alpha L$−th layer, in the distilled model. In this subsection, we consider the normalized output of an infinitely deep model whose $\alpha L$-th layer is hidden and subjected to the attack. The output of the distilled model is

$$\hat{f}_\infty(\mathbf{X}) = \lim_{L \to \infty} \frac{f(\mathbf{X}; \boldsymbol{\theta}_{\text{dist}}(\{\alpha L\}))}{\|f(\mathbf{X}; \boldsymbol{\theta}_{\text{dist}}(\{\alpha L\}))\|_F},$$

where $\|\cdot\|_F$ denotes the Frobenius norm. We consider an infinitely deep network as the ideal model, reflecting the sufficient depth of most large-scale models in practice. The following theorem establishes the existence of a critical value $\alpha^*$ such that if $\alpha < \alpha^*$, the output matrix of the distilled LLM has rank one. Conversely, if $\alpha > \alpha^*$, the output matrix has rank strictly greater than one.

**Theorem 1.** *Assume that $\mathbb{P}_{\mathbf{X} \times Y}$ is defined on a countable domain $\mathcal{X} \times \mathcal{Y}$ with $\mathbf{0}_{n \times d} \notin \mathcal{X}$. Assume that parameter matrices $\{K_i, Q_i\}_{i \geq 1}$ in the victim model $f$ have uniform bounded norms, i.e., $\|K_i\| \leq D$ and $\|Q_i\| \leq D$ for some $D > 0$. There exists an $\alpha^* \in (0, 1)$ depending on $D$ such that the following two statements are true.*

*(1) If $\alpha < \alpha^*$ and $\{K_i, Q_i\}_{i \geq 1}$ are parameter matrices of the victim model, with $\hat{K}_{\alpha L}$ and $\hat{Q}_{\alpha L}$ as distilled parameters drawn from a continuous distribution on $\mathbb{R}^{n \times d}$, the matrix $\hat{f}_\infty(\mathbf{X})$ almost surely has rank one for all inputs $\mathbf{X}$.*

*(2) If $\alpha > \alpha^*$, there exists a victim model with parameter sequence $\{K_i, Q_i\}_{i \geq 1}$ such that for any distilled parameters $\hat{K}_{\alpha L}$ and $\hat{Q}_{\alpha L}$, the matrix $\hat{f}_\infty(\mathbf{X})$ has rank greater than one for some $\mathbf{X}$.*

**Remark 1:** The proof is provided in Appendix A. This theorem demonstrates that if the distilled parameters of the bottom layers (i.e., $\alpha < \alpha^*$) are obtained through a randomized algorithm, such as stochastic gradient descent, with a continuous distribution supported on $\mathbb{R}^{n \times d}$, the distillation will certainly fail, as the feature matrix degenerate. In contrast, keeping the later layers secured (i.e., $\alpha > \alpha^*$) does not maintain this property, indicating that it is more effective to secure the bottom layers before the transition layer, rather than the later ones.

**Remark 2:** The existence of $\hat{f}_\infty(\mathbf{X})$ is a non-trivial result. While the mapping $\varphi_i$ admits a fixed point at $\mathbf{X} = \mathbf{0}_{n \times d}$, the convergence of the iterative process governed by $\varphi_i$ cannot be guaranteed using the contraction mapping theorem, as $\varphi_i$ does not satisfy the contraction property for any pair $(Q_i, K_i)$. This complexity becomes particularly evident in the special case where $n = 1$ and $\mathbf{X}$ is a column vector. Here, the output of $\varphi_i$ satisfies the relation $\langle \mathbf{1}_d, \varphi_i(\mathbf{X}; K_i, Q_i) \rangle = 2\langle \mathbf{1}_d, \mathbf{X} \rangle$, implying that the iteration diverges unless $\mathbf{X}$ is orthogonal to $\mathbf{1}_d$. However, the divergence is not arbitrary; rather, the theorem reveals that it occurs in a fixed, well-defined direction. This insight ensures the existence of a normalized output, which remains stable and meaningful despite the lack of strict convergence.

**Remark 3:** The existence of $\alpha^* \in (0, 1)$ is also a non-trivial statement, as $\alpha^*$ could potentially be zero, which would imply the absence of a critical layer such that securing layers prior to it guarantees the failure of the recovered model's functionality. The primary challenge lies in demonstrating that perturbations to the earlier layers result in rank-one outputs, a property that does not universally hold for arbitrary perturbations. To address this, we establish an alternative result: given an input matrix $\mathbf{X}$, rank-one outputs can be guaranteed if the perturbation matrices $K_i$ and $Q_i$ are chosen to avoid specific zero-measure sets, denoted as $\mathcal{K}(\mathbf{X})$ and $\mathcal{Q}(\mathbf{X})$, respectively. Assuming a countable domain $\mathcal{X} \times \mathcal{Y}$, which is typical for structured inputs such as sentences or images, it follows that the perturbation matrices to be avoided belong to the countable union of these sets, defined as $\mathcal{K} = \bigcup_{\mathbf{X} \in \mathcal{X}} \mathcal{K}(\mathbf{X})$ and $\mathcal{Q} = \bigcup_{\mathbf{X} \in \mathcal{X}} \mathcal{Q}(\mathbf{X})$. Since this union remains a zero-measure set, avoiding these specific sets ensures that the conditions of the theorem are satisfied for any input matrix $\mathbf{X}$.

Theorem 1 shows that securing bottom layers improves security. Next, we propose SOLID, a preliminary LLM deployment framework solution that balance model protection and customization.

## 4.2 SOLID: SEMI-OPEN LOCAL INFRASTRUCTURE DEPLOYMENT FRAMEWORK

We propose a method to approximately find the smallest bottom layer index set $I$ that satisfies $R(I) \leq (1 + \varepsilon)R([L])$ for any small $\varepsilon > 0$. A simple approach is to start with $I_l = \{1, \dots, l\}$ for each $l$ beginning from 1, then evaluate the distillation ratio $R(I_l)$ after the attack, and identify the smallest $l$ that meets the inequality. This extensive fine-tuning process is time-consuming, prompting the critical question: *Can we create a fine-tuning-free metric that predicts LLM performance under model distillation attacks?* Hence, our goal is to establish a metric directly correlated with the distillation ratio.

In the distillation ratio $R(I)$, each $I$ has the same denominator, so our focus is on a metric related to the numerator, specifically $\mathbb{E}[s(f(\mathbf{X}; \boldsymbol{\theta}_{\text{FT}}(I, \mathcal{D})), Y)]$, which measures the average performance score of the distilled model. This average performance score generally inversely correlates with the average testing loss with the expression $L(\boldsymbol{\theta}) \triangleq \mathbb{E}_{\mathbf{X} \times Y}[\ell(f(\mathbf{X}; \boldsymbol{\theta}), Y)]$, where $\ell$ denotes the cross-entropy loss employed by LLM. Hence, we aim at finding the smallest $I$ such that

$$L(\boldsymbol{\theta}_{\text{dist}}(I, \mathcal{D})) \geq (1 - \varepsilon)L(\boldsymbol{\theta}_{\text{dist}}([L], \mathcal{D})).$$

However, calculating both sides of this inequality requires knowing the distilled parameters from the fine-tuning process. To bypass this, we aim for an approximate solution. The distilled parameters are generated through gradient descent, starting from the initial parameters $\boldsymbol{\theta}_0(I)$, with the hidden layers being randomly initialized. Using the Taylor Expansion, we find

$$L(\boldsymbol{\theta}_{\text{dist}}(I, \mathcal{D})) = L(\boldsymbol{\theta}_0(I, \mathcal{D})) + \mathcal{O}(\mathbb{E}\|\boldsymbol{\theta}_{\text{dist}}(I, \mathcal{D}) - \boldsymbol{\theta}_0(I)\|_2).$$

Previous research (Choi et al., 2024; Bailly et al., 2022) indicates that the difference $\|\boldsymbol{\theta}_{\text{dist}}(I, \mathcal{D}) - \boldsymbol{\theta}_0(I)\|_2$ is minimal in large networks compared to the dataset size $|\mathcal{D}|$. In models such as single-layer ReLU networks (Anthony et al., 1999; Zou et al., 2020), this difference scales as $\mathcal{O}\left(\frac{|\mathcal{D}|}{\sqrt{N}}\right)$ (Jacot et al., 2018; Wei et al., 2019), where $N$, the number of model parameters, far exceeds the dataset size in large language models (LLMs) (Dubey et al., 2024; Liu et al., 2024). The first term, independent of fine-tuning, dominates and effectively predicts the distillation ratio. We refer to this term as the **Distillation Difficulty** (DD($I$)), defined as

$$\text{DD}(I) = \mathbb{E}[L(\boldsymbol{\theta}_0(I))].$$

This score, which can be estimated using a sample average, represents the distilled model performance of the model when specific layers $I$ are secured. A higher **DD**($I$) suggests better security performance, indicating a lower distillation ratio $R(I)$. Therefore, our SOLID operates in the following way. SOLID begins by sampling evaluation data targeting general capabilities from the underlying distribution, and then computes DD($I_l$) for each set of secured layers $I_l = \{1, ..., l\}$ for $l = 1, ..., L$. SOLID stops at the smallest $l^*$ that satisfies DD($I_{l^*}$) $\geq (1 - \varepsilon)$DD($[L]$).

## 5 EXPERIMENTS

In this section, we conduct experiments to answer the following research questions:

- **RQ1.** Can query-based distillation attacks distill the functionality of the entire model under the baseline deployment that secures the top layer?
- **RQ2.** How do secured layer location and amount affect the security-customization trade-off?
- **RQ3.** Does securing bottom layers (SOLID) offer a better balance between model theft risk and customization performance compared to baseline deployments?

## 5.1 EXPERIMENTAL SETTINGS

We begin by introducing our experimental setups. Details can be found in Appendix B.

**Models.** We consider **five** open-source, decoder-only structured LLMs with various architectures. Specifically, we select Llama2-70B-chat, Llama2-7B-chat (Touvron et al., 2023), Mistral-7B-v0.1 (Jiang et al., 2023), Phi-2 (Abdin et al., 2024), and Phi-1.5 (Li et al., 2023). We designate these pre-trained models as the base models for adaptation and victims in model distillation attacks.

**Attack Methods.** We distill models produced by different protection approaches using three attack methods: FT-all, FT-closed and SEM. Following (He et al., 2021), a diverse attack set is required

for full distillation. Therefore, we merge data evenly form two general datasets, MMLU benchmark (Hendrycks et al., 2021) and Alpaca 52k (Wang et al., 2022), resulting in a 51k combined set. Additionally, we build four larger general datasets (100k–500k) to strengthen the attack.

**Baselines.** We compare SOLID with three baselines: SAP-DP, the fully-secured approach (Eiras et al., 2024), and DarkneTZ (Mo et al., 2020). The SAP (Shen et al., 2023) framework exposes the first six decoder layers and secures the rest. SAP-DP extends SAP by adding Laplace noise to model outputs to enhance protection (Lee et al., 2018). The fully-secured approach represents the extreme, securing all layers for maximal security, while DarkneTZ protects only the final decoder layer.

**Implementation Details of SOLID.** We apply the SOLID algorithm to identify the smallest secure set $I$ such that $R(I) \leq (1 + \varepsilon)R([L])$. To calculate distillation difficulty (DD), we use cross-entropy loss and approximate the expectation over samples distributed on the general domain and randomly initialized secured parameters. This is done using a 1,500-sample evaluation set randomly sampled from the MMLU benchmark and Alpaca 52k, with secured parameters initialized via Xavier initialization and averaged over three random seeds (20, 42, 1234). In our experiments, we find that $\varepsilon = 0.05$ yields optimal performance.

**Evaluation Benchmarks** We assess the model adaptability on six downstream tasks: Code (Zheng et al., 2024b), Math (Yue et al., 2023), Medical (Zhang et al., 2023), Finance (Wang et al., 2023b), Law (Guha et al., 2024), and Alignment (Meng et al., 2024). To fully evaluate recovered functionalities, we focus on six capabilities domains following Llama2 report (Touvron et al., 2023). Specifically, we assess the recovered model across **sixteen** benchmarks grouped into (1) *Commonsense Reasoning* (Rsn.); (2) *Reading Comprehension* (Read.); (3) *World Knowledge* (Knl.); (4) *Code*; (5) *Math*; and (6) *General Ability* (Gen.).

**Metrics.** We measure customization through model's improvements on benchmarks. For security, we calculate the "Average Distillation Ratio" (ADR) by averaging the distillation ratios across benchmarks. A lower ADR indicates higher security offered by the secure set.

## 5.2 FAILURE IN DEFENSE (RQ1)

We evaluate security of DarkneTZ using three distillation strategies. Based on the results shown in Tables 1 and 2, we have following observations.

**Obs1: DarkneTZ, which secures only the last decoder layer, fails to protect the model against all three attacks.** As shown in Table 1, DarkneTZ achieves ADRs generally exceeding 73%. Notably, on Llama2-7B, it surpasses 100% distillation ratio on the MMLU and BBH datasets, indicating that the distilled model outperforms the original on these tasks. Similarly, Table 2 highlights consistent failure patterns against FT-closed and SEM attacks, with DarkneTZ maintaining ADRs above 75%, demonstrating the ability of these strategies to recover significant model functionality.

## 5.3 SECURITY-CUSTOMIZATION TRADE-OFF (RQ2)

We conduct two experiments to analyze the impact of secured layer placement and quantity on the trade-off between security and customization. First, we secure one layer in Llama2-7B and two in Phi-2, varying their placement. Second, we incrementally secure both models by adding protected layers, starting from the smallest module (`k_project`) of the first decoder layer. These models are evaluated under the FT-all distillation attack and customized for the math domain. The results, as shown in Figure 4, lead to the following observations.

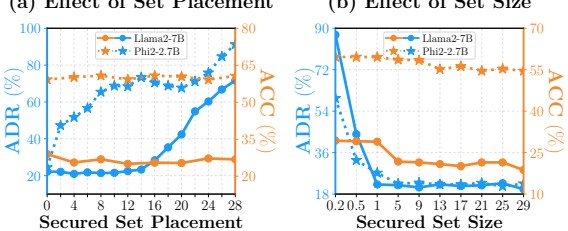

Figure 4: (a) shows the trade-off between security and customization for Llama2-7B and Phi-2 with different placements of same-sized secured sets. (b) shows the trade-off as the secured set size increases from the first decoder layer. Smaller ADR indicates higher security and higher ACC reflects better customizability.

**Observation 2: Secured layer placement significantly impacts security, consistent with Theorem 1, but has small effect on customization performance.** As shown in Figure 4(a), for

Table 1: Distillation ratios on 6 functionalities under FT-all (SOLID|SAP-DP|Fully-secured|DarkneTZ). "H.E." in Code domain presents the benchmark "HumanEval".

| | Benchmark | Llama2-70B | Llama2-7B | Mistral-7B | Phi-2 |
|---|---|---|---|---|---|
| **Rsn.** | PIQA | 62.6\|59.8\|63.0\|99.3 | 64.7\|64.7\|64.6\|99.1 | 63.0\|61.2\|60.2\|92.2 | 68.3\|65.6\|65.7\|99.1 |
| | Winogrande | 68.5\|67.7\|68.3\|98.3 | 76.8\|74.8\|76.6\|100. | 67.2\|69.0\|68.3\|89.5 | 68.3\|64.9\|64.8\|99.1 |
| | ARC-easy | 31.9\|32.8\|31.3\|98.5 | 36.3\|35.5\|34.9\|97.6 | 32.3\|34.7\|32.0\|86.6 | 43.2\|35.3\|33.9\|99.5 |
| | ARC-challenge | 38.5\|38.1\|44.2\|99.2 | 47.8\|46.6\|50.9\|100. | 39.7\|42.6\|44.5\|81.4 | 36.8\|36.6\|35.3\|99.5 |
| | Hellaswag | 31.4\|31.4\|32.4\|98.1 | 33.9\|34.0\|35.0\|96.6 | 32.2\|32.0\|31.3\|84.6 | 37.4\|37.3\|34.3\|96.5 |
| **Read.** | LAMBADA | 0.01\|0.00\|0.00\|88.6 | 0.02\|0.00\|0.01\|92.2 | 0.16\|0.00\|0.01\|67.9 | 1.34\|0.04\|0.00\|94.6 |
| | BoolQ | 47.2\|47.1\|53.9\|100. | 59.5\|56.0\|65.0\|99.6 | 48.3\|46.8\|56.7\|97.3 | 56.7\|50.3\|55.8\|100. |
| | SQuADv2 | 1.50\|1.68\|0.34\|55.3 | 0.68\|0.88\|0.82\|59.5 | 1.69\|0.36\|0.93\|50.7 | 3.65\|0.39\|0.90\|62.9 |
| | OBQA | 54.5\|54.5\|57.1\|99.6 | 57.4\|52.5\|59.2\|94.8 | 57.7\|56.8\|56.3\|84.0 | 0.00\|0.00\|0.02\|94.3 |
| **Knl.** | NaturalQuestions | 0.00\|0.02\|0.00\|40.1 | 0.01\|0.01\|0.08\|53.6 | 0.00\|0.00\|0.02\|31.8 | 0.01\|0.00\|0.06\|87.4 |
| | TriviaQA | 0.00\|0.02\|0.00\|72.3 | 0.00\|0.00\|0.03\|73.8 | 0.00\|0.00\|0.01\|38.7 | 0.01\|0.00\|0.01\|68.9 |
| **Code** | MBPP&H.E. | 0.00\|0.00\|0.00\|58.6 | 0.00\|0.00\|0.00\|90.9 | 0.00\|0.00\|0.00\|40.2 | 0.00\|0.00\|0.00\|91.1 |
| **Math** | GSM8K | 0.02\|0.00\|0.06\|79.6 | 0.00\|0.00\|0.00\|78.6 | 0.00\|0.00\|0.00\|31.1 | 0.00\|0.00\|0.00\|86.2 |
| **Gen.** | MMLU | 36.8\|38.3\|36.5\|96.7 | 52.9\|50.0\|53.3\|110. | 40.4\|36.9\|37.2\|81.7 | 42.6\|40.3\|40.5\|99.5 |
| | BBH | 0.00\|0.00\|0.00\|93.3 | 0.00\|0.00\|0.00\|101. | 0.00\|0.00\|0.00\|63.3 | 0.01\|0.00\|0.00\|94.8 |
| **Avg. Distil. Ratio(↓)** | | **21.9**\|21.8\|22.8\|**77.9** | **25.3**\|24.4\|25.9\|**86.5** | **22.5**\|22.4\|22.8\|**73.7** | **23.9**\|22.3\|22.4\|**88.9** |
| **Secured Ratio(↓)** | | **2.50**\|92.5\|100.\|**1.25** | **3.16**\|81.3\|100.\|**3.16** | **3.16**\|81.3\|100.\|**3.16** | **6.25**\|81.3\|100.\|**3.16** |

Llama2-7B, security transitions at the fourteenth layer, with ADR consistently near 20% for earlier sets, indicating stronger security than protecting later layers. Meanwhile, customization accuracy remains stable across placements, highlighting the advantage of securing pre-transition layers. In contrast, Phi-2 transitions earlier at the first layer set, where only the first set balances security and customization, with later sets reducing security. These results suggest that securing layers before the transition layer optimizes the security-customization trade-off. More results are in Appendix B.7.

**Observation 3: Increasing the number of secured layers enhances security but reduces customization.** As shown in Figure 4(b), the ADR of Llama2-7B decreases from 85% to 22% after securing an entire decoder layer, indicating improved security. However, customization accuracy drops from 29% to 21% as the number of secured layers increases from one to five, reflecting reduced customization flexibility. A similar trend is observed in Phi-2, suggesting that while increasing the number of secured layers enhances security (lower ADR), it negatively impacts customization flexibility (lower ACC) in both models. Further details are in Appendix B.8.

## 5.4 EFFECTIVENESS OF SOLID (RQ3)

We compare the security of SOLID with baseline deployments across three distillation strategies. The results lead to the following observations.

**Observation 4: SOLID offers comparable security against model distillation to the highest level of protection (fully-secured), while securing significantly fewer parameters.** As shown in Table 1, SOLID achieves a similar security level (ADR) to SAP-DP and the fully-secured approach across four architectures and various domains, while securing at most 6.25% of parameters, compared to at least 80% for the others. For example, on Llama2-70B, SOLID secures only 1.25% of parameters yet achieves an ADR of 21.9%,

Table 2: Distillation ratios of Llama2-70B under FT-closed and SEM attacks.

| Strat. | Method | Rsn. | Read. | Knl. | C.&M. | Gen. | ADR |
|---|---|---|---|---|---|---|---|
| **FT-c.** | SOLID | 47.1 | 21.6 | 0.00 | 0.03 | 18.7 | 22.6 |
| | SAP-DP | 46.2 | 19.5 | 0.00 | 0.00 | 19.0 | 21.8 |
| | F-Secured | 47.8 | 21.2 | 0.00 | 0.08 | 18.5 | 22.8 |
| | DarkneTZ | 98.7 | 69.3 | 58.3 | 65.9 | 95.0 | 78.1 |
| **SEM** | SOLID | 48.2 | 21.9 | 0.00 | 0.00 | 18.5 | 22.4 |
| | SAP-DP | 47.1 | 21.1 | 0.00 | 0.00 | 18.3 | 22.3 |
| | F-Secured | 47.8 | 21.2 | 0.00 | 0.08 | 18.5 | 22.8 |
| | DarkneTZ | 98.8 | 71.2 | 54.2 | 66.3 | 94.1 | 77.4 |

comparable to SAP-DP (21.8%) and the fully-secured approach (22.8%), which protect 92.5% and 100% of parameters, respectively. Furthermore, under FT-closed and SEM attacks, SOLID also matches the security level provided by SAP-DP and the fully-secured approach. Table 2 shows that under FT-closed attack, the ADR differences between SOLID, SAP-DP, and the fully-secured approach remain below 2.1% across six domains. Similarly, under SEM attack, the distillation ratios

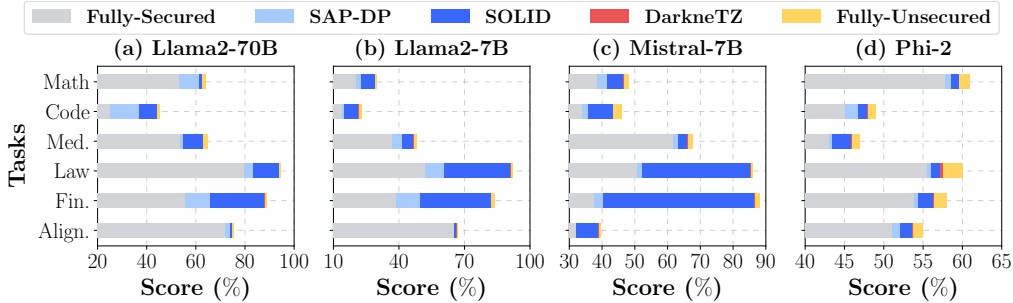

Figure 5: Customization performance comparison of secured models on six downstream tasks.

closely aligned with the other two approaches. These results confirm that SOLID effectively protects against distillation attacks while securing significantly fewer parameters.

**Observation 5: The security of SOLID cannot be easily compromised by simply increasing the dataset scale.** As shown in Table 3, the distillation ratios for SOLID increase marginally with larger datasets, showing only a 0.5% ADR rise when scaling from 51k to 500k samples. In contrast, DarkneTZ exhibits a significant increase in the ADR, from $86.5\%$ to $96.9\%$, over the same dataset size range. This highlights the robustness of SOLID's security against increasing attack dataset sizes. Details of the attack datasets are provided in Appendix B.2.

Table 3: SOLID vs. Dataset scales. ADR-Da. represents the ADR by DarkneTZ.

| Scale | Rsn. | Read. | Knl. | C.&M. | Gen. | ADR | ADR-Da. |
|---|---|---|---|---|---|---|---|
| 51k | 51.7 | 21.6 | 0.01 | 0.00 | 28.3 | 25.3 | 86.5 |
| 100k | 51.3 | 21.5 | 0.13 | 0.00 | 29.6 | 25.3 | 89.1 |
| 200k | 51.4 | 21.7 | 0.11 | 0.00 | 29.7 | 25.2 | 91.3 |
| 300k | 51.6 | 21.7 | 0.11 | 0.00 | 30.5 | 25.5 | 94.5 |
| 500k | 51.8 | 22.0 | 0.09 | 0.00 | 30.8 | 25.8 | 96.9 |

**Observation 6: SOLID consistently outperforms baseline deployments in customization while achieving security levels comparable to fully-secured approaches. Its customization performance approaches the flexibility of full parameter fine-tuning.** As the results shown in Figure 5, we observe that, in the Law domain, SOLID improves scores by 10% over SAP-DP and fully-secured approach on Llama2-70B, with a 35% improvement on the 7B models. Similar gains are observed on Phi-2, though the improvement in Law reduces to 1%. Additionally, the performance of SOLID consistently matches the performance of full parameter fine-tuning across four architectures, with differences within 4%. This demonstrates that securing a small portion of parameters minimally impacts customization while providing strong protection against distillation attacks. Further results can be found in Appendix B.6.

We summarize the security and customization performance of each deployment in Figure 6. SOLID achieves an optimal balance between distillation prevention and customization, outperforming other baselines. In the next subsection, we discuss how the distillation difficulty metric optimizes the security-customization trade-off.

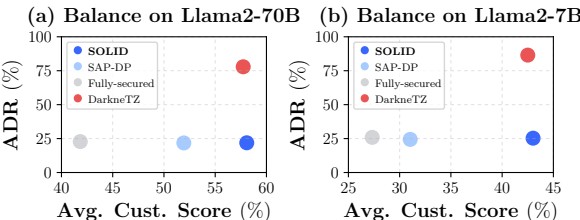

Figure 6: ADRs vs. average customization score. Points closer to the bottom-right indicate better balance.

## 6 CONCLUSION

In this paper, we advocate for a balanced approach to on-premises LLM deployment, addressing the dilemma between data privacy and model confidentiality. Building on our theorem, we propose SOLID, a simple yet effective semi-open solution that efficiently secures a few bottom layers of LLMs. We believe that our work shifts the view from skepticism to opportunity, paving a viable middle path for secure AI deployment that protects user privacy, safeguards vendor intellectual property, and preserves model customization flexibility. While our position may invite both support and critique, we hope it provides meaningful insights and fosters ongoing discourse. If this paper contributes to constructive engagement within the community, it will have fulfilled its purpose.

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

## A PROOF OF THEOREM 1

In this section, we prove Theorem 1. We first revisit the our model, present several important lemmas and finally present the proof.

### A.1 MODEL OVERVIEW

The distilled model $f(\mathbf{X}; \boldsymbol{\theta})$ is structured as a sequence of $L$ transformer layers,

$$f(\mathbf{X}) = \varphi_L \circ \varphi_{L-1} \circ \dots \circ \varphi_{\alpha L+1} \circ \hat{\varphi}_{\alpha L} \circ_{\alpha L-1} \circ \dots \circ \varphi_1(\mathbf{X}), \tag{2}$$

where $\mathbf{X} \in \mathbb{R}^{n \times d}$ represents the input, interpreted as an assembly of $n$ tokens, each possessing $d$ hidden dimensions. Each transformer layer, indexed by $1 \leq i \leq L$, is represented by $\varphi_i$, which maps $\mathbb{R}^{n \times d}$ to $\mathbb{R}^{n \times d}$ and can be defined as follows,

$$\varphi_i\left(\mathbf{X}; K_i, Q_i\right) = \left[\mathbf{I}_n + \text{softmax}\left(\frac{\mathbf{X}Q_i(\mathbf{X}K_i)^\top}{\sqrt{d_Q}\|\mathbf{X}\|^2}\right)\right]\mathbf{X}, \tag{3}$$

where $Q_i \in \mathbb{R}^{d \times d_Q}$, $K_i \in \mathbb{R}^{d \times d_Q}$ represent projection parameter matrices. Here, the $\alpha L$-th layer is the distilled layer and the others are the public layers. For simplicity, we use the function $\hat{\varphi}_{\alpha L}$ to denote mapping of the distilled layer, i.e., $\hat{\varphi}_{\alpha L}(\mathbf{X}) = \varphi_{\alpha L}(\mathbf{X}; \hat{K}_{\alpha L}, \hat{Q}_{\alpha L})$.

### A.2 BOUNDS ON DIFFERENT ORTHOGONAL COMPONENTS

**Lemma 1.** *For any $1 \leq l \leq L$, $1 \leq p \leq d$, any $\mathbf{X} \in \mathbb{R}^{n \times d}$, we have*

$$\max_{\boldsymbol{v}:\|\boldsymbol{v}\|_2=1,\boldsymbol{v}\perp\mathbb{I}_n} \left|\boldsymbol{v}^\top \varphi_l\left(\mathbf{X}; K_l, Q_l\right)[p]\right| \leq (1 + \beta_D) \max_{\boldsymbol{v}:\|\boldsymbol{v}\|_2=1,\boldsymbol{v}\perp\mathbb{I}_n} \left|\boldsymbol{v}^\top \mathbf{X}[p]\right|, \tag{4}$$

*where $\mathbb{I}_n$ is a column vector with dimensions $n \times 1$ and each element is 1, $\mathbf{X}[p]$ is the $p$-th column of the input $\mathbf{X}$, $\varphi_l\left(\mathbf{X}; K_l, Q_l\right)[p]$ is the $p$-th column of the $l$-th self-attention output, the coefficient $\beta_D$ satisfies $0 < \beta_D < 1$ and it is related to the upper bound of the L2-norm of matrices $K_l, Q_l$.*

*Proof.* Let $\boldsymbol{u} = \left\{\boldsymbol{u}_{l,1} = \frac{\mathbb{I}_n}{\sqrt{n}}, \boldsymbol{u}_{l,2}, \dots, \boldsymbol{u}_{l,n}\right\}$ denote the eigenvectors of $\text{softmax}\left(\frac{\mathbf{X}Q_l(\mathbf{X}K_l)^\top}{\sqrt{d_Q}\|\mathbf{X}\|^2}\right)$.

Assume $\sigma_{l,1}, \sigma_{l,2}, \dots, \sigma_{l,n}$ denote the eigenvalues of $\text{softmax}\left(\frac{\mathbf{X}Q_i(\mathbf{X}K_i)^\top}{\sqrt{d_Q}\|\mathbf{X}\|^2}\right)$ and $-1 < \sigma_{l,n} < \beta_D$ for any $l, n$. Thus we have

$$\boldsymbol{v}^\top \varphi_l\left(\mathbf{X}; K_l, Q_l\right)[p] = \boldsymbol{v}^\top \left[\mathbf{I}_n + \text{softmax}\left(\frac{\mathbf{X}Q_l(\mathbf{X}K_l)^\top}{\sqrt{d_Q}\|\mathbf{X}\|^2}\right)\right]\mathbf{X}[p] \tag{5a}$$

$$= \boldsymbol{v}^\top \left[\mathbf{I}_n + \text{softmax}\left(\frac{\mathbf{X}Q_l(\mathbf{X}K_l)^\top}{\sqrt{d_Q}\|\mathbf{X}\|^2}\right)\right]\sum_{k=1}^{n} \alpha_{pk}\boldsymbol{u}_{l,k} \tag{5b}$$

$$= \boldsymbol{v}^\top \sum_{k=1}^{n} \alpha_{pk}(1 + \sigma_{l,k})\boldsymbol{u}_{l,k} \tag{5c}$$

$$\leq \max_{\boldsymbol{v}:\|\boldsymbol{v}\|_2=1,\boldsymbol{v}\perp\mathbb{I}_n} \left|\sum_{k=2}^{n} \alpha_{pk}(1 + \sigma_{l,k})\boldsymbol{v}^\top u_{l,k}\right| \tag{5d}$$

$$= \left\|\sum_{k=2}^{n} \alpha_{pk}(1 + \sigma_{l,k})\boldsymbol{u}_{l,k}\right\|_2 \tag{5e}$$

$$= \left[\sum_{k=2}^{n} \alpha_{pk}^2(1 + \sigma_{l,k})^2\right]^{1/2} \tag{5f}$$

$$\leq (1 + \beta_D) \max_{\boldsymbol{v}:\|\boldsymbol{v}\|_2=1,\boldsymbol{v}\perp\mathbb{I}_n} \left|\boldsymbol{v}^\top \mathbf{X}[p]\right|, \tag{5g}$$

where

$$\beta_D = \max_{\|K_l\|_2 \leq D, \|Q_l\|_2 \leq D} \max_{\boldsymbol{v}: \|\boldsymbol{v}\|_2 = 1, \boldsymbol{v} \perp \mathbb{I}_n} \left\| \text{softmax}\left(\frac{\mathbf{X}Q_l(\mathbf{X}K_l)^\top}{\sqrt{d_Q}\|\mathbf{X}\|^2}\right)\boldsymbol{v} \right\|_2 < 1.$$

The equation equation 5c is due to $\boldsymbol{u}_{l,k}$ are the eigenvectors of $\text{softmax}\left(\frac{\mathbf{X}Q_l(\mathbf{X}K_l)^\top}{\sqrt{d_Q}\|\mathbf{X}\|^2}\right)$. The inequality equation 5e is because when $\boldsymbol{v} = \frac{\sum_{k=2}^n \alpha_{pk}(1+\sigma_{l,k})\boldsymbol{u}_{l,k}}{\left\|\sum_{k=2}^n \alpha_{pk}(1+\sigma_{l,k})u_{l,k}\right\|_2}$, we have the maximum value.

$\square$

**Lemma 2.** *For any $K_l, Q_l \in \mathbb{R}^{d \times s}$ and any $\mathbf{X} \in \mathbb{R}^{n \times d}$, the following equation always holds:*

$$\left|\mathbb{I}_n^\top \varphi_i(\mathbf{X}; K_i, Q_i)[p]\right| = 2\left|\mathbb{I}_n^\top \mathbf{X}[p]\right|, \tag{6}$$

*where $\mathbf{X}[p]$ is the $p$-th column of the input $\mathbf{X}$, $\varphi_i(\mathbf{X}; K_i, Q_i)[p]$ is the $p$-th column of the $l$-th self-attention output.*

*Proof.* Assume that a set of orthogonal basis for $\mathbb{R}^n$ is $\{\boldsymbol{u_1}, \boldsymbol{u_2}, \ldots, \boldsymbol{u_n}\}$, where $\boldsymbol{u_1} = \frac{\mathbb{I}_n}{\sqrt{n}}$. Then we can rewrite $\mathbf{X}[p]$ as $\mathbf{X}[p] = \sum_{j=1}^n \alpha_{pj}\boldsymbol{u_j}$, where $\alpha_{pj}(1 \leq p \leq d)$ are the corresponding coefficients for the $p$-th column of $\mathbf{X}$ under the orthogonal basis. Next, we calculate $\left|\mathbb{I}_n^\top f(\mathbf{X})[p]\right|$ and $\left|\mathbb{I}_n^\top \mathbf{X}[p]\right|$, respectively. Note that $\mathbb{I}_n^\top \boldsymbol{u_j} = 0$ for all $j \neq 1$. Therefore, we can obtain that,

$$\mathbb{I}_n^\top \mathbf{X}[p] = \sqrt{n}\alpha_{p1}. \tag{7}$$

Then we can get

$$\left|\mathbb{I}_n^\top \mathbf{X}[p]\right| = |\sqrt{n}\alpha_{p1}|. \tag{8}$$

Let $\sigma_{i1}, \sigma_{i2}, \ldots, \sigma_{in}$ denote the eigenvalues of $\text{softmax}\left(\frac{\mathbf{X}Q_i(\mathbf{X}K_i)^\top}{\sqrt{d_Q}\|\mathbf{X}\|^2}\right)$. Applying the Perron–Frobenius theorem for Markov matrices (Lemmens & Nussbaum, 2012), we deduce that for the matrix $\text{softmax}\left(\frac{\mathbf{X}Q_l(\mathbf{X}K_i)^\top}{\sqrt{d_Q}\|\mathbf{X}\|^2}\right)$, there exists only one eigenvalue equal to 1, while all other eigenvalues in absolute value are strictly less than 1. Without loss of generality, we assume $\sigma_{i1} = 1$, implying $|\sigma_{ij}| < 1$ for $j \neq 1$. Recalling the definition of $\varphi_i(\mathbf{X}; K_i, Q_i)$ and considering the linear operation, we can rewrite it as follows:

$$\varphi_i(\mathbf{X}; K_i, Q_i)[p] = \sum_{j=1}^n \alpha_{pj}(1 + \sigma_{ij})\boldsymbol{u_j}. \tag{9}$$

Then we calculate the term $\left|\mathbb{I}_n^\top \varphi_i(\mathbf{X}; K_i, Q_i)[p]\right|$ as follows,

$$\left|\mathbb{I}_n^\top \varphi_i(\mathbf{X}; K_i, Q_i)[p]\right| = \left|\mathbb{I}_n^\top (\sum_{j=1}^n \alpha_{pj}(1 + \sigma_{ij})\boldsymbol{u_j}\right| \tag{10a}$$

$$= \left|\sqrt{n}(\alpha_{p1}(1 + \sigma_{i1}))\right| \tag{10b}$$

$$= 2|\sqrt{n}\alpha_{p1}|, \tag{10c}$$

where equation 10a is induced by substituting the equation equation 9 into $\left|\mathbb{I}_n^\top \varphi_i(\mathbf{X}; K_i, Q_i)[p]\right|$, equation 10b is due to $\mathbb{I}_n^\top \boldsymbol{u_j} = 0$ for all $j \neq 1$, equation 10c follows the fact that $\sigma_{i1} = 1$.

$\square$

### A.3 PROOF OF THEOREM 1

We first prove the following result. For simplicity of notations, we use $f(\mathbf{X})[p]$ to denote the $p$-th $(1 \leq p \leq d)$ column of the the distilled model $f(\mathbf{X})$, where the parameters in the $\alpha L$-th layer is replaced with the matrices $\hat{K}_{\alpha L}$ and $\hat{Q}_{\alpha L}$. We use the function $\hat{\varphi}_{\alpha L}(\mathbf{X}) = \varphi_{\alpha L}(\mathbf{X}; \hat{K}_{\alpha L}, \hat{Q}_{\alpha L})$ to

denote the mapping of the $(\alpha L)$-th layer. Then we are going to show that there exists $\alpha^\star = \log_2 \frac{2}{1+\beta_D}$ and $0 < \beta_D < 1$ makes the following equations hold.

(1) Assume $\alpha < \alpha^\star$. For any $\mathbf{X}$, $\|K_i\|_2 \leq D$, $\|Q_i\|_2 \leq D$, there exists a zero measure set $\mathcal{K}(\mathbf{X})$ and $\mathcal{Q}(\mathbf{X})$ such that

$$\lim_{L\to\infty} \left\| \frac{f(\mathbf{X})[p]}{\|f(\mathbf{X})[p]\|_2} - \frac{\mathbb{I}_n}{\sqrt{n}} \right\|_2 = 0. \tag{11}$$

(2) For any $\alpha > \alpha^\star$, there exists a sequence of matrix $\{K_i, Q_i\}_{i\geq 1}$ such that for any distilled matrix $K_{\alpha L}$ and $Q_{\alpha L}$, we have $\|K_i\|_2 \leq D$, $\|Q_i\|_2 \leq D$, we have,

$$\lim_{L\to\infty} \left\| \frac{f(\mathbf{X})[p]}{\|f(\mathbf{X})[p]\|_2} - \frac{\mathbb{I}_n}{\sqrt{n}} \right\|_2 = \sqrt{2}. \tag{12}$$

*Proof.* Based on Lemma equation 1, we obtain that

$$\max_{\boldsymbol{v}:\|\boldsymbol{v}\|_2=1,\boldsymbol{v}\perp\mathbb{I}_n} \left| \boldsymbol{v}^\top f(\mathbf{X})[p] \right| \leq (1+\beta)^L \max_{\boldsymbol{v}:\|\boldsymbol{v}\|_2=1,\boldsymbol{v}\perp\mathbb{I}_n} \left| \boldsymbol{v}^\top \mathbf{X}[p] \right|. \tag{13}$$

Based on Lemma equation 2, we know that

$$\left| \mathbb{I}_n^\top f(\mathbf{X})[p] \right| = 2^{(1-\alpha)L-1} \left| \mathbb{I}_n^\top \hat{\varphi}_{\alpha L} \circ \varphi_{\alpha L-1} \circ \cdots \circ \varphi_1(\mathbf{X})[p] \right|. \tag{14}$$

We firstly prove the equation equation 11. When

$$\left| \mathbb{I}_n^\top f(\mathbf{X})[p] \right| = 2^{(1-\alpha)L-1} \left| \mathbb{I}_n^\top \hat{\varphi}_{\alpha L} \circ \varphi_{\alpha L-1} \circ \cdots \circ \varphi_1(\mathbf{X})[p] \right| \neq 0, \tag{15}$$

then we have

$$\left\| \frac{f(\mathbf{X})[p]}{\|f(\mathbf{X})[p]\|_2} - \frac{\mathbb{I}_n}{\sqrt{n}} \right\|_2 = \left[ 2 - \frac{2\mathbb{I}_n^\top f(\mathbf{X})[p]}{\sqrt{n}\sqrt{\frac{(\mathbb{I}_n^\top f(\mathbf{X})[p])^2}{n} + (\boldsymbol{v}^\top f(\mathbf{X})[p])^2}}} \right]^{1/2} \tag{16a}$$

$$= \sqrt{2} \left[ 1 - \frac{1}{\sqrt{1 + \frac{n(\boldsymbol{v}^\top f(\mathbf{X})[p])^2}{(\mathbb{I}_n^\top f(\mathbf{X})[p])^2}}}} \right]^{1/2} \tag{16b}$$

$$\leq \sqrt{2} \left[ 1 - \frac{1}{\sqrt{1 + \frac{n(1+\beta)^{2L}|\boldsymbol{v}^\top \mathbf{X}[p]|^2}{2^{2[(1-\alpha)L-1]}|\mathbb{I}_n^\top \hat{\varphi}_{\alpha L} \circ \varphi_{\alpha L-1} \circ \cdots \circ \varphi_1(\mathbf{X})[p]|^2}}}} \right]^{1/2} \tag{16c}$$

$$\leq 2\sqrt{2n} \left( \frac{1+\beta}{2^{1-\alpha}} \right)^L \frac{\left| \boldsymbol{v}^\top \mathbf{X}[p] \right|}{\left| \mathbb{I}_n^\top \hat{\varphi}_{\alpha L} \circ \varphi_{\alpha L-1} \circ \cdots \circ \varphi_1(\mathbf{X})[p] \right|}, \tag{16d}$$

where the inequality equation 16c is based on the inequality equation 13 and equation 14. The inequality equation 16d is based on Lemma equation 3. Therefore, if $\alpha < \log_2 \frac{2}{1+\beta_D}$ and $\left| \mathbb{I}_n^\top f(\mathbf{X})[p] \right| \neq 0$, then we have $\lim_{L\to\infty} \left( \frac{1+\beta_D}{2^{1-\alpha}} \right)^L = 0$. Now we can consider when $\left| \mathbb{I}_n^\top f(\mathbf{X})[p] \right| = 0$. In fact, it is easy to show that this can only happens when $\hat{K}_{\alpha L}$ and $\hat{Q}_{\alpha L}$ belong to certain sets making $\left| \mathbb{I}_n^\top f(\mathbf{X})[p] \right| = 0$, which corresponds to zero measure set $\mathcal{K}(\mathbf{X})$ and $\mathcal{Q}(\mathbf{X})$ depending on the input $\mathbf{X}$. Since the input space is countable, therefore, the union $\cup_{\mathbf{X}\in\mathcal{X}}\mathcal{K}(\mathbf{X})$ and $\cup_{\mathbf{X}\in\mathcal{X}}\mathcal{Q}(\mathbf{X})$ are also zero-measure sets.

To prove equation equation 12, let $K^\star$, $Q^\star$ with $\|K^\star\|_2 \leq D$, $\|Q^\star\|_2 \leq D$ satisfy the following condition,

$$\max_{\boldsymbol{v}:\|\boldsymbol{v}\|_2=1,\boldsymbol{v}\perp\mathbb{I}_n} \left\| \text{softmax}\left( \frac{\mathbf{X}Q_l(\mathbf{X}K_l)^\top}{\sqrt{d_Q}\|\mathbf{X}\|^2} \right) \boldsymbol{v} \right\|_2 = \beta_D. \tag{17}$$

Let $\boldsymbol{v}^\star$ be the solver of the above optimization problem equation 17 and consider the $K_l = K^\star$, $Q_l = Q^\star$ and $\mathbf{X}^\star = [\boldsymbol{v}^\star, \boldsymbol{v}^\star, \cdots, \boldsymbol{v}^\star]$. Clearly, $\boldsymbol{v}^\star \perp \mathbb{I}_n$. Assume there exists $\boldsymbol{u} : \|\boldsymbol{u}^\star\|_2 = 1$ satisfying $\boldsymbol{u}^\star \perp \mathbb{I}_n$, $\boldsymbol{u}^\star \perp \boldsymbol{v}^\star$, therefore we can rewrite $f(\mathbf{X}^\star)[p]$ as follows,

$$f(\mathbf{X}^\star)[p] = \frac{\mathbb{I}_n^\top}{\sqrt{n}} f(\mathbf{X}^\star) \frac{\mathbb{I}_n}{\sqrt{n}} + \boldsymbol{v}^{\star\top} f(\mathbf{X}^\star)\boldsymbol{v}^\star + \boldsymbol{u}^{\star\top} f(\mathbf{X}^\star)\boldsymbol{u}^\star. \tag{18}$$

For any $1 \leq l \leq L$, based on Lemma equation 1, we know that

$$\left| \boldsymbol{v}^{*\top} f\left(\mathbf{X}^{\star}\right)[p] \right| = (1 + \beta_D)^L \left| \boldsymbol{v}^{*\top} \mathbf{X}^{\star}[p] \right|. \tag{19}$$

Since

$$\left| \mathbb{I}_n^\top f\left(\mathbf{X}^{\star}\right)[p] \right| = 2^L \left| \mathbb{I}_n^\top \mathbf{X}^{\star}[p] \right| = \left| \mathbb{I}_n^\top \boldsymbol{v}^{\star} \right| = 0 \tag{20}$$

and

$$\left| \boldsymbol{v}^{*\top} f\left(\mathbf{X}^{\star}\right)[p] \right| = (1 + \beta_D)^L \left| \boldsymbol{v}^{*\top} \mathbf{X}^{\star}[p] \right| \neq 0, \tag{21}$$

then we have

$$\left\| \frac{f(\mathbf{X}^{\star})[p]}{\|f(\mathbf{X}^{\star})[p]\|_2} - \frac{\mathbb{I}_n}{\sqrt{n}} \right\|_2 = \left[ 2 - \frac{2\mathbb{I}_n^\top f(\mathbf{X}^{\star})[p]}{\sqrt{n}\|f(\mathbf{X}^{\star})[p]\|_2} \right]^{1/2} \tag{22a}$$

$$= \left[ 2 - \frac{2\mathbb{I}_n^\top}{\sqrt{n}} \frac{f(\mathbf{X}^{\star})[p]}{\sqrt{\frac{1}{n}(\mathbb{I}_n^\top f(\mathbf{X}^{\star})[p])^2 + (\boldsymbol{v}^{\star\top} f(\mathbf{X}^{\star})[p])^2 + (\boldsymbol{u}^{\star\top} f(\mathbf{X}^{\star})[p])^2}} \right]^{1/2} \tag{22b}$$

$$\geq \left[ 2 - \frac{2\mathbb{I}_n^\top}{\sqrt{n}} \frac{f(\mathbf{X}^{\star})[p]}{\sqrt{\frac{1}{n}(\mathbb{I}_n^\top f(\mathbf{X}^{\star})[p])^2 + (\boldsymbol{v}^{\star\top} f(\mathbf{X}^{\star})[p])^2}} \right]^{1/2} \tag{22c}$$

$$= \left[ 2 - 2\frac{\frac{\mathbb{I}_n^\top f(\mathbf{X}^{\star})[p]}{\sqrt{n}|\boldsymbol{v}^{\star\top} f(\mathbf{X}^{\star})[p]|}}{\sqrt{1 + \frac{|\mathbb{I}_n^\top f(\mathbf{X}^{\star})[p]|^2}{n|\boldsymbol{v}^{\star\top} f(\mathbf{X}^{\star})[p])|^2}}} \right]^{1/2} \tag{22d}$$

$$= \left[ 2 - 2\frac{\frac{2^{(1-\alpha)L-1}|\mathbb{I}_n^\top \hat{\varphi}_{\alpha L} \circ \varphi_{\alpha L-1} \circ \cdots \circ \varphi_1(\mathbf{X}^{\star})[p]|}{\sqrt{n}(1+\beta_D)^L|\boldsymbol{v}^{\star\top} \mathbf{X}^{\star}[p]|}}{\sqrt{1 + \frac{2^{2[(1-\alpha)L-1]}}{n(1+\beta_D)^{2L}} \frac{|\mathbb{I}_n^\top \hat{\varphi}_{\alpha L} \circ \varphi_{\alpha L-1} \circ \cdots \circ \varphi_1(\mathbf{X}^{\star})[p]|^2}{|\boldsymbol{v}^{\star\top} \mathbf{X}^{\star}[p]|^2}}} \right]^{1/2}, \tag{22e}$$

where equation equation 22b is based on equation 18, equation equation 22e is based on equation 21 and equation 14. When $\alpha > \log_2 \frac{2}{1+\beta_D}$, we have $\lim_{L \to \infty} \left( \frac{2^{1-\alpha}}{1+\beta_D} \right)^L = 0$. Thus we have $\lim_{L \to \infty} \left\| \frac{f(\mathbf{X}^{\star})[p]}{\|f(\mathbf{X}^{\star}[p]\|_2} - \frac{\mathbb{I}_n}{\sqrt{n}} \right\|_2 = \sqrt{2}$. This indicates that the $p$-th column of the output matrix $f(\mathbf{X}^{\star})$ is not parallel to $\mathbf{I}_n$ for any $p$. This further indicates that the output matrix does not have the identical vector in each row. $\qquad\square$

## A.4 TECHNICAL LEMMA

**Lemma 3.** *For any $x \in (0, 1)$, it always holds $\left[ 1 - \frac{1}{\sqrt{1+x^2}} \right]^{1/2} \leq x$.*

*Proof.* To establish the inequality $\left[ 1 - \frac{1}{\sqrt{1+x^2}} \right]^{1/2} \leq x$, we begin by proving,

$$1 - \frac{1}{\sqrt{1 + x^2}} \leq x^2. \tag{23}$$

To demonstrate equation 23, we equivalently show

$$1 - x^2 \leq \frac{1}{\sqrt{1 + x^2}}. \tag{24}$$

Subsequently, it suffices to verify

$$(1 - x^2)(\sqrt{1 + x^2}) \leq 1. \tag{25}$$

This is equivalent to proving

$$(1 - x^2)^2(1 + x^2) \leq 1. \tag{26}$$

Thus, our focus shifts to demonstrating

$$(1 - x^2)(1 - x^4) \leq 1. \tag{27}$$

Clearly, equation 27 holds true for any $x \in (0, 1)$. $\qquad\square$

# B    EXPERIMENT DETAILS

To more intuitively compare the security differences between the SOLID method and a fully-secured approach, we define $\Delta\mathbf{ADR}(I) = \mathrm{ADR}(I) - \mathrm{ADR}([L])$ to assess the resilience of the secured set $I$ relative to the fully-secured approach. A smaller value of $\Delta\mathbf{ADR}$ indicates resilience similar to that of the fully-secured model.

## B.1    MODEL DETAILS.

The foundation models we use in our experiments are selected from open-source repositories, and Table 4 shows the basic information of the models and their sources. Specifically, we employ Llama2-70B-chat[1], Llama2-7B-chat[2], and Mistral-7B-v0.1[3]. For smaller models, we select Phi-2[4] and Phi-1.5[5]. We also consider OPT model[6], which has only 350 million parameters and 24 decoder layers.

Table 4: Model Info

| Model | Size | Decoder Layers |
|-------|------|----------------|
| Llama2-70B-chat (Touvron et al., 2023) | 70B | 80 |
| Llama2-7B-chat (Touvron et al., 2023) | 7B | 32 |
| Mistral-7B-v0.1 (Jiang et al., 2023) | 7B | 32 |
| Phi-2 (Abdin et al., 2024) | 2.7B | 32 |
| Phi-1.5 (Li et al., 2023) | 1.3B | 24 |
| OPT (Zhang et al., 2022) | 350M | 24 |

## B.2    DISTILLATION ATTACKS.

**Attack implementation details.** In performing FT-all and FT-secure model distillation attacks, we adhere to the training hyper-parameters outlined in the Llama2 report (Touvron et al., 2023), employing the AdamW optimizer with a cosine learning rate scheduler. The initial learning rate is set to $2 \times 10^{-5}$, with a weight decay of 0.1, a batch size of 128, and bfloat16 precision for input sequences of 512 tokens. The LLaMA2-70B model is trained for 3 epochs with a random seed of 42, while other models are trained for 5 epochs across three seeds: 42, 1234, and 20. Despite limiting training to 3 epochs for the 70B model, the training loss stabilized effectively. Our implementation builds upon the llama-recipes repository provided by META.

For SEM attacks, distinct configurations were employed for SOLID and SAP-DP. In the case of SOLID, hidden representations from the secure-source components were collected and paired with the input data to train a substitute model. In contrast, for SAP-DP, representations from the sixth decoder layer and the model's final logits were utilized to construct the training dataset. In accordance with (Tamber et al., 2024), we applied a learning rate of 1.5e-4, a weight decay of 0.01, and a linear learning rate scheduler with 500 warmup steps. Both training and validation batch sizes were set to 32, with MSE as the loss function. SOLID was trained for 30 epochs due to its smaller model size, whereas SAP-DP was trained for 5 epochs.

All distillation experiments were conducted on Nvidia 4090 24G, 6000 Ada 48G, and A100 80G GPUs, utilizing PyTorch 2.2.0 and CUDA 11.8 on Ubuntu 20.04.6 LTS.

---

[1]https://huggingface.co/meta-llama/Llama-2-70b-chat-hf

[2]https://huggingface.co/meta-llama/Llama-2-7b-chat-hf

[3]https://huggingface.co/mistralai/Mistral-7B-v0.1

[4]https://huggingface.co/microsoft/phi-2

[5]https://huggingface.co/microsoft/phi-1_5

[6]https://huggingface.co/facebook/opt-350m

**Base 51k Distillation Dataset.** We ensure dataset coverage and reliability by using a 1:1 ratio of the MMLU auxiliary training set [7] and Alpaca dataset [8], extracting 25.5k samples from each. From the MMLU auxiliary training data (Hendrycks et al., 2021), we sample 50%, and from Alpaca (Taori et al., 2023), we use a step size of 2 to enhance diversity. The datasets are then formatted for model training, applying Alpaca and MMLU prompts from Table 5.

Table 5: Prompts for Alpaca and MMLU auxiliary training data

| Dataset | Prompt Type | Description |
|---------|-------------|-------------|
| **Alpaca** | with input | Below is an instruction that describes a task, paired with an input that provides further context. Write a response that appropriately completes the request. |
| | w/o input | Below is an instruction that describes a task. Write a response that appropriately completes the request. |
| **MMLU** | Question Answering | Below is a question with no choices. Write the correct answer that appropriately solves the question. |
| | Multiple Choice | The following is a multiple choice question, paired with choices. Answer the question in the format: "Choice:content". |

**Extra Distillation Datasets.** To enhance dataset diversity, the 100K, 200K, 300K, and 500K datasets integrate additional specialized sources. As detailed in Table 6, these sources include Baize (Xu et al., 2023) (158K English multi-turn conversations via ChatGPT's self-chat), MathInstruct (Yue et al., 2023) (260K curated math instruction instances focusing on hybrid reasoning), and OpenOrca (Mukherjee et al., 2023) (augmented FLAN collection with 1M GPT-4 completions and 3.2M GPT-3.5 completions). These enrichments are intended to support complex computational and theoretical tasks, offering broader topic coverage.

Table 6: Composition of variously sized datasets

| Raw Data Set | 51k | 100k | 200k | 300k | 500k |
|--------------|-----|------|------|------|------|
| Alpaca | 25.5 | 50 | 40 | 50 | 50 |
| MMLU auxiliary training set | 25.5 | 50 | 40 | 100 | 100 |
| Baize-MedQuAD | 0 | 0 | 40 | 50 | 50 |
| Baize-Quora | 0 | 0 | 40 | 50 | 50 |
| Baize-Stackoverflow | 0 | 0 | 40 | 50 | 50 |
| MathInstruct | 0 | 0 | 4 | 6 | 20 |
| OpenOrca | 0 | 0 | 0 | 0 | 180 |

**Validation Datasets.** Table 7 outlines the composition of the validation datasets. For *Validation Dataset 1*, we extracted 50% from each of the 57 MMLU validation sub-datasets, totaling 1.5K instances, paired with Alpaca data selected using a step size of 751. This dataset is used with the 51K and 100K training sets. For larger training sets (200K, 300K, and 500K), *Validation Dataset 2* was created by adding 400 instances from three Baize subsets, expanding the validation set to 4.0K.

### B.3 BASELINES.

In this section, we provide further details on the baselines used in our comparisons: SAP-DP and fully-secured. These schemes represent different strategies, each with distinct trade-offs in terms of customizability and security against model distillation attacks.

---

[7] https://github.com/hendrycks/test

[8] https://github.com/tatsu-lab/stanford_alpaca/blob/main/alpaca_data.json

Table 7: Composition of validation datasets of different sizes

| Raw Data Set | Validation Set | Evaluation Set |
|---|---|---|
| Alpaca | 765 | 765 |
| MMLU auxiliary training set | 751 | 751 |
| Baize-MedQuAD | 0 | 850 |
| Baize-Quora | 0 | 850 |
| Baize-Stackoverflow | 0 | 850 |
| **Total Length** | 1516 | 4066 |

**SAP.** The Split-and-Privatize (SAP) framework (Shen et al., 2023) offers an approach to balance between protecting model privacy and data privacy while maintaining competitive performance. Specifically, the SAP framework keeps the bottom six encoder layers open, allowing user access and fine-tuning while securing the deeper layers on the vendor.

**SAP-DP.** To further strengthen protection while maintaining competitive performance, we extend SAP by incorporating differential privacy techniques by adding Laplace noise to perturb the logits during the fine-tuning process (Lee et al., 2018). The Laplace Distribution with mean $\mu$ and scale $b$ is the distribution with probability density function:

$$\text{Laplace}(x|\mu, b) = \frac{1}{2b} \exp\left(-\frac{|x - \mu|}{b}\right)$$

Specifically, in SAP-DP, the noise $n$ is sampled: $n \sim \text{Laplace}(0, 0.5)$ and added to the output logits of the model to balance privacy protection and model performance.

**Fully-secured.** Following (Eiras et al., 2024), we use the fully-secured approach as a baseline. This assumes the adversary has no access to internal model parameters, treating the model as a black-box, where only output data can be collected. We slightly broaden this setup by assuming the adversary knows the model's architecture but no other details. Thus, distilling the fully-secured model involves using the collected data to retrain a model with the same architecture to restore its general functionality.

**DarkneTZ.** Based on the work of (Mo et al., 2020), we use DarkneTZ as a baseline to test whether protecting only the output layers is sufficient to defend against distillation attacks. In this setup, we assume the adversary has no access to the model parameters of the output layers, specifically the last decoder layer. Similar to the SAP framework, this approach allows the adversary to access and fine-tune all layers except the final decoder layer.

### B.4    IMPLEMENTATION DETAILS OF SOLID.

**Evaluation Datasets.** We created a 1.5K Evaluation Set to assess model security under various secure-sourcing strategies. This set includes 50% of entries from each of the 57 MMLU validation sub-datasets (Hendrycks et al., 2021), distinct from Validation Set outlined in Table 7. Additionally, we selected an equal number of Alpaca dataset (Taori et al., 2023), using a step size of 751, ensuring no overlap with the Validation Set.

**Hyper-parameter Sensitivity.** As shown in Figure 7, we evaluate SOLID's sensitivity to tolerance magnitude $\varepsilon$, adjusting it from 0.05 to 1 in 0.05 increments while calculating the $\Delta$ADR for six distilled models. The results indicate that SOLID is minimally sensitive to changes in $\varepsilon$, with $\Delta$ADR values stabilizing as $\varepsilon$ increases. This stability arises from the need for a smaller secured layer at higher $\varepsilon$, allowing the condition $R(I) \leq (1 + \varepsilon)R([L])$ to be met with fewer layers. Additionally, the increase in $\Delta$ADR is smaller for larger models, suggesting that privatizing more parameters beyond a certain point offers diminishing returns in security.

### B.5    EVALUATION BENCHMARKS

Most of our evaluations are conducted using the lm-evaluation suite (Gao et al., 2023), the bigcode-evaluation-harness platform (Ben Allal et al., 2022), and MT-Bench (Zheng et al., 2023). For specific

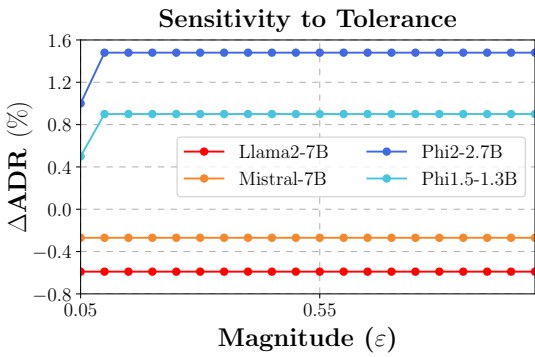

Figure 7: Sensitivity on $\varepsilon$.

domains, such as finance and law, we utilize the official benchmark testing codes provided by their respective communities, as detailed below.

**Evaluation on Customizabilities.** We assess the customizability of models across six domains, as detailed in Table 8. Each domain includes specific benchmarks and metrics designed to evaluate different aspects of the model's performance in relation to customizability. In particular, for evaluating medical capabilities, we select two subcategories from the MMLU benchmark that are related to the medical domain: *mmlu_anatomy* and *mmlu_professional_medicine*. For assessing legal reasoning, we select 10 multiple-choice and judgment-based subcategories from Legalbench. The performance of the model in these legal tasks is measured using perplexity, following the prompt structure provided by Legalbench. Specifically, the selected subcategories include: *cuad_audit_rights*, *canada_tax_court_outcomes*, *definition_classification*, *cuad_affiliate_license-licensee*, *learned_hands_business*, *contract_nli_survival_of_obligations*, *contract_nli_explicit_identification*, *contract_nli_confidentiality_of_agreement*, *hearsay*, and *contract_qa*.

Table 8: Details of the Six Customizability Benchmarks

| Domain | Benchmark | Metric | n-shot | Reference |
|---|---|---|---|---|
| Code | HumanEval | Pass@1 | 0 | (Chen et al., 2021) |
| | MBPP | Pass@1 | 1 | (Austin et al., 2021) |
| Math | GSM8K | Exact Match | 8 | (Cobbe et al., 2021) |
| Medical | MMLU_Medical | Accuracy | 5 | (Hendrycks et al., 2021) |
| Finance | FPB | F1 | 0 | (Wang et al., 2023a) |
| Law | LegalBench | Accuracy | 0 | (Guha et al., 2023) |
| Alignment | MT-Bench | Score | (GPT-4) | (Zheng et al., 2023) |

**Evaluation on Security.** We follow the Llama-2 report (Touvron et al., 2023) to evaluate the distilled model, including 16 benchmarks, which are categorized into 6 groups. Table 9 summarizes the functionality benchmarks used in our experiments, along with their test methods and performance metrics. Our model ranks choices in multiple-choice tasks and generates answers for open-ended generation tasks.

### B.6 MODEL CUSTOMIZATION

**Datasets.** To fine-tune the models for domain-specific tasks, we utilized several datasets tailored to different sectors, including Code (Zheng et al., 2024b), Math (Yue et al., 2023), Medical (Zhang et al., 2023), Finance (Wang et al., 2023b), Law (Guha et al., 2024), and Alignment (Meng et al., 2024). Table 10 lists the customization training datasets used in the experiments. For the code domain,

Table 9: Details of the Sixteen Functionality Benchmarks

| Domain | Benchmark | Metric | n-shot | Reference |
|--------|-----------|--------|--------|-----------|
| **Commonsense Reasoning** | PIQA | Accuracy | 0 | (Bisk et al., 2020) |
| | Hellaswag | Accuracy | 0 | (Zellers et al., 2019) |
| | Winogrande | Accuracy | 0 | (Sakaguchi et al., 2019) |
| | ARC_easy | Accuracy | 0 | (Clark et al., 2018) |
| | ARC_challenge | Accuracy | 0 | (Clark et al., 2018) |
| **Reading Comprehension** | OpenBookQ | Accuracy | 0 | (Mihaylov et al., 2018) |
| | LAMBADA | Accuracy | 0 | (Paperno et al., 2016) |
| | BoolQ | Accuracy | 0 | (Clark et al., 2019) |
| | SQuADv2 | HasAns_EM | 2 | (Rajpurkar et al., 2018) |
| | SQuADv2 | HasAns_F1 | 2 | (Rajpurkar et al., 2018) |
| **World Knowledge** | NaturalQuestions | Exact Match | 5 | (Kwiatkowski et al., 2019) |
| | TriviaQA | Exact Match | 5 | (Joshi et al., 2017) |
| **Code** | HumanEval | Pass@1 | 0 | (Chen et al., 2021) |
| | MBPP | Pass@1 | 1 | (Austin et al., 2021) |
| **Math** | GSM8K | Exact Match | 8 | (Cobbe et al., 2021) |
| **General Ability** | MMLU | Accuracy | 5 | (Hendrycks et al., 2021) |
| | BBH | Accuracy | 3 | (Suzgun et al., 2022) |

we combine the datasets from CodeFeedback and CodeAlpaca. For law and finance, we merge all training datasets from Legalbench and FinGPT respectively. These datasets are then prepared for model training using the Alpaca prompts outlined in Table 5. Additionally, we randomly select 3,000 samples to serve as the validation dataset.

Table 10: Customization Training Datasets Composition

| Domain | Dataset Name | Size | Reference |
|--------|--------------|------|-----------|
| **Code** | CodeFeedback | 156k | (Zheng et al., 2024a) |
| | CodeAlpaca | 20k | (Chaudhary, 2023) |
| **Math** | MathInstruction | 262K | (Yue et al., 2023) |
| **Medical** | MedMCQA | 183k | (Zhang et al., 2023) |
| **Law** | Legalbench | 90k | (Guha et al., 2023) |
| **Finance** | FinGPT | 204k | (Wang et al., 2023a) |
| **Alignment** | Ultrafeedback | 62k | (Cui et al., 2024) |

**Customization Training Hyperparameters.** In model customization, we use different hyperparameters depending on the model size. For LLaMA2-70B, we apply QLoRA with the settings outlined in Table 11, while for 7B models, we use LoRA. For smaller models like Phi2 and Phi-1.5, we fine-tune all model parameters. For LLaMA2-70B, we fine-tune it as a quantized 4-bit model over 1 epoch, starting with a learning rate of $1.5 \times 10^{-6}$. For the 7B models, we train for 3 epochs, with a seed value of 42. The training setup includes a weight decay of 0.1, a batch size of 128, a warmup ratio of 0.03, and input sequences of 512 tokens, following standard experimental practices (Hu et al., 2021). For Phi2 and Phi-1.5, we use the training hyperparameters from the LLaMA2 report. We employ the AdamW optimizer with a cosine learning rate scheduler, starting with a learning rate of $2 \times 10^{-5}$, a

weight decay of 0.1, a batch size of 128, and use bfloat16 precision for 512-token input sequences. Specifically, for alignment, we follow SimPO (Meng et al., 2024) and set the preference parameters $\beta = 2$ and $\gamma = 1$. The learning rate is $1 \times 10^{-6}$ for LLaMA2-70B and $5 \times 10^{-7}$ for the 7B and smaller models. All experiments are conducted using the LLaMA-Factory on Nvidia 4090 24G, 6000 Ada 48G, and A100 80G GPUs, with PyTorch 2.2.0 and CUDA 11.8 on Ubuntu 20.04.6 LTS.

Table 11: The Hyperparameters for Customization Training.

| Model | Method | Rank $r$ | Lora $\alpha$ | Dropout | Learning Rate | Epochs | Warmup R. |
|---|---|---|---|---|---|---|---|
| **Llama2-70B** | QLoRA | 96 | 16 | 0.05 | 1.50E-04 | 1 | 0.03 |
| **Llama2-7B** | LoRA | 32 | 64 | 0.05 | 2.00E-05 | 3 | 0.03 |
| **Mistral-7B** | LoRA | 32 | 64 | 0.05 | 1.00E-06 | 3 | 0.03 |

### B.7 SECURITY AND CUSTOMIZATION TRANSITIONS

For the LLaMA2-7B model, the smallest secure-source layer set identified by SOLID consists of a single decoder layer, whereas for Phi-2, it includes two decoder layers. Consequently, for LLaMA2-7B, we opted to secure-source each even-indexed layer, while for Phi-2, we chose to secure-source non-overlapping pairs of layers (e.g., layers 0-1, 2-3). For each selected layer set, we first secure-source them, then subjected the semi-open model to FT-all attacks, and subsequently calculated the $\Delta$ADR of the layer set to assess its security.

When verifying the customization transition, due to computational constraints, we validated only every other layer set for both models (e.g., secure-source layers 0, 0-4, 0-8 . . . ). Specifically, we applied LoRA-based customization on LLaMA2-7B in the math domain, while for Phi-2, we utilized the full finetuning approach. The experimental hyperparameters remain consistent with those outlined in the Appendix B.6.

We further computed the $\Delta$ADR for each secure-source set within Mistral-7B-v0.1 and Phi-1.5. In these models, the smallest secure-source set identified by SOLID consists of one decoder layer and two decoder layers, respectively. Following the same experimental configuration as LLaMA2-7B and Phi-2, we secured each even-indexed layer for Mistral-7B, and non-overlapping pairs of layers for Phi-1.5. The complete results demonstrating the transition layers within the Mistral-7B and Phi-1.5 model that secure two non-overlapping consecutive layers are depicted in Figure 8. Once again, we observed a distinct presence of transition layers. Specifically, in Mistral-7B, the transition layer appears at the 24th layer, while in Phi-1.5, it is located within the first layer set.

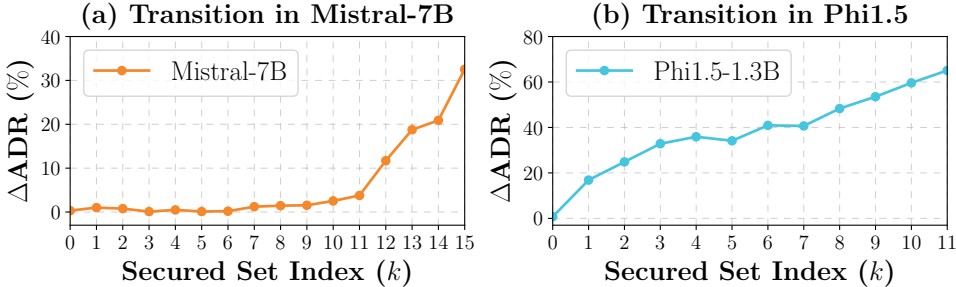

Figure 8: Security changes in Miatral-7B and Phi-1.5.

### B.8 SECURITY ACROSS SECURE SIZES

To examine the influence of Secure layer size on model security, we conduct experiments on Secure-sourcing different amounts and proportions of parameters in the model's decoder layer. We give instructions on the detailed setting of secured models in Table 12. The module names are all derived from the overall implementation functions of each model in the Transformers open-source repositories in Table 4. We utilize abbreviated module names to denote specific settings.

Table 12: Secured Sizes Setting. "*" indicates an entire decoder layer.

| | | Llama-7B | Mistral-7B | Phi2-2.7B | Phi1.5-1.3B |
|---|---|---|---|---|---|
| **Proportion** | 0.25% | $W_k$ | $W_q, W_k$ | $W_k$ | $W_k$ |
| | 0.50% | $W_q, W_k$ | $W_o, MLP_{up}$ | $W_q, W_k$ | $W_q, W_k$ |
| | 1% | $W_q, W_k, W_v, W_o$ | $W_q, W_k, W_v, W_o$ | $W_q, W_k, W_v, W_d$ | $W_q, W_k, , W_v$ |
| | 3% | 0 | 0 | 0 | 0 |
| | 7% | 0-1 | 0-1 | 0-1 | 0-1 |
| | 15% | 0-4 | 0-4 | 0-3 | 0-3 |
| | 30% | 0-9 | 0-9 | 0-9 | 0-6 |
| | 50% | 0-15 | 0-15 | 0-15,$W_{em}$ | 0-11,$W_{em}$ |
| | 100% | Fully-secured | Fully-secured | Fully-secured | Fully-secured |
| **Quantity** | 20M | $W_k$ | $W_q, W_k$ | $W_q, W_k, W_v$ | $W_q, W_k, W_v, W_d$ |
| | 50M | $W_q, W_k, W_v$ | $W_q, W_k, W_v, W_o$ | $MLP$ | 0 |
| | 100M | $W_q, W_k, W_v, MLP$ | $W_q, W_k, W_v, W_o, MLP$ | $0, W_q, W_k, W_v$ | 0-1 |
| | 160M | $W_q, W_k, W_v, W_o, MLP$ | $W_q, W_k, W_v, W_o, MLP$ | 0-1 | 0-2 |
| | 200M | 0 | 0 | 0-1, $W_q, W_k, W_v, W_d, MLP_{f1}$ | 0-3 |
| | 300M | $0, W_q, W_v, W_o, MLP_{up}$ | $0, W_q, W_v, W_o, MLP_{up}$ | 0-3 | 0-5 |
| | 600M | 0-2 | 0-2 | 0-7 | 0-11 |

We further computed $\Delta$ADR by close-sourcing varying quantities and proportions of parameters under FT-all attacks on three additional models. As shown in Figure 9 and Figure 10(a), we observed the same pattern as with Llama2-7B, where security emerges once a sufficient number of parameters are secured. For example, on Mistral-7B, security occurs after secure-sourcing 100 million parameters, which is less than a single decoder layer. Secure-sourcing fewer parameters leads to a notable drop in security, with $\Delta$ADR rising to around 40%. Beyond this threshold, security stabilizes near 0% $\Delta$ADR. This pattern holds across all models, highlighting a critical threshold for effective secure-source. Furthermore, different architectures require varying secure-sourcing quantities to achieve security, even with similar model sizes. For instance, Mistral-7B reaches security by secure-sourcing 100 million parameters, Llama2-7B requires 200 million, and Phi-1.5 needs a higher rate of 7%, compared to 3% for Llama2-7B.

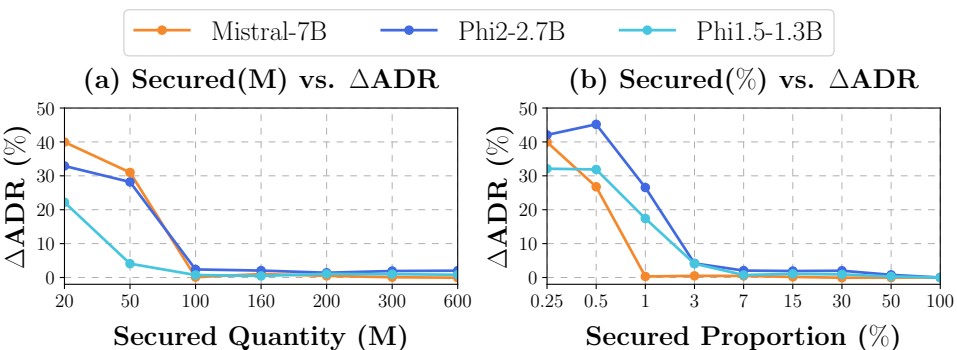

Figure 9: $\Delta$ ADR for different secure parameter quantities and proportions.

We explore how secured parameter ratio impacts the model security in Llama2-7B, as shown in Figure 10(b). For instance, technical skills such as Math show earlier transitions, with security emerging at 1% parameters secured, whereas domains such as Commonsense Reasoning require hiding 3%. In summary, secure-sourcing a small portion of parameters can provide sufficient security against model distillation, meanwhile, technical capabilities tend to be more challenging to distill than other domains.

## B.9 EFFECTIVENESS OF DISTILLATION DIFFICULTY

The complete Pearson and Spearman results are presented in Table 13, revealing a negative correlation between RS and the average distillation ratio. For example, in Llama2-7B, both Pearson and Spearman coefficients fall below -0.77. Similar trends are seen in models with varying architectures and sizes,

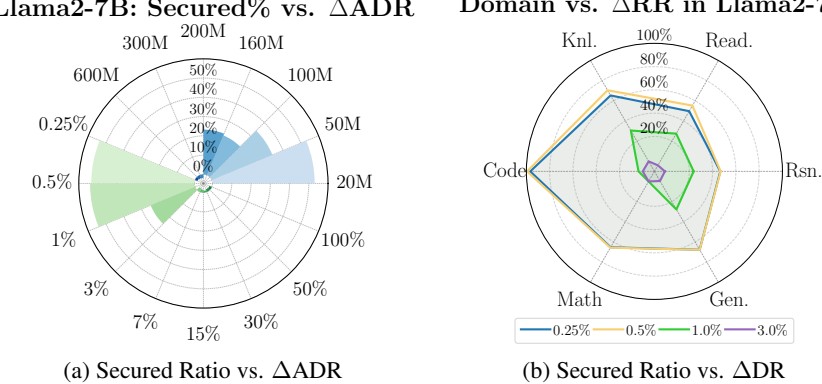

(a) Secured Ratio vs. $\Delta$ADR

(b) Secured Ratio vs. $\Delta$DR

Figure 10: $\Delta$ADR and $\Delta$DR changes in Llama2-7B with varying secured parameter ratios.

confirming that RD is a reliable predictor of distilled model performance and demonstrating the effectiveness of SOLID. Additionally, Figure 11 shows scatter plots depicting the relationship between $\Delta$ADR and Distillation Difficulty($\uparrow$)s across four models, along with the corresponding Pearson and Spearman correlation coefficients. The Distillation Difficulty($\uparrow$)s were obtained from Section 5.3. As illustrated in Figure 11, we observe a clear trend: an increase in $\Delta$ADR corresponds to a decrease in model scores across all models analyzed. This inverse relationship is consistently supported by strong negative values for both Pearson and Spearman correlation coefficients, with the most significant negative correlation seen in Phi2-2.7B, indicating a substantial drop in model scores as $\Delta$ADR increases.

Table 13: Correlation coefficients (Spearman | Pearson) between distillation ratio and distillation difficult.

| Model | Rsn. | Read. | Knl. | Code & Math | Gen. | Avg. |
|---|---|---|---|---|---|---|
| Llama2-7B | -0.83 \| -0.97 | -0.77 \| -0.96 | -0.83 \| -0.95 | -0.85 \| -0.90 | -0.82 \| -0.93 | -0.80 \| -0.98 |
| Mistral-7B | -0.83 \| -0.89 | -0.82 \| -0.91 | -0.82 \| -0.94 | -0.78 \| -0.95 | -0.76 \| -0.87 | -0.87 \| -0.92 |
| Phi-2 | -0.93 \| -0.96 | -0.84 \| -0.96 | -0.84 \| -0.87 | -0.84 \| -0.80 | -0.84 \| -0.84 | -0.87 \| -0.95 |
| Phi-1.5 | -0.86 \| -0.97 | -0.78 \| -0.94 | -0.83 \| -0.94 | -0.90 \| -0.80 | -0.84 \| -0.89 | -0.80 \| -0.94 |

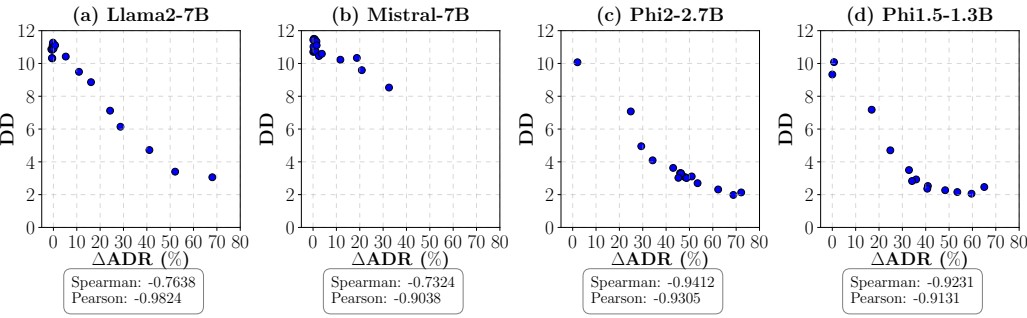

Figure 11: Correlation Analysis of $\Delta$ADR and Distillation Difficulty Across Different Models.

