# OpenReview forum: "On-Premises LLM Deployment Demands a Middle Path: Preserving Privacy Without Sacrificing Model Confidentiality"
_ICLR.cc/2025/Workshop/BuildingTrust — BuildingTrust_

### Official Review · Reviewer_iPZD · 2025-02-28
**A Semi-Open Framework for Secure Model Customization**

**Rating:** 8
**Confidence:** 4

**Review:**

# Review
This paper authors propose a semi-open deployment framework that protects certain key layers (typically the bottom layers) of the model to prevent distillation attacks while allowing fine-tuning of the unprotected parts. The framework introduces a novel metric called Distillation Difficulty (DD) to evaluate the security of the model under different protection strategies.

## Strengths:
1. New Framework: The paper proposes a semi-open deployment framework that offers a middle ground between fully closed-source APIs and open-weight models, addressing both privacy and misuse risks.
2. Transition layer and DD: The paper proposeds transition layer , which is used to determine whether specific layers should be protected.And proposes Distillation Difficulty (DD) metric to evaluate the security of the model under specific protection strategies.
3. Experiments: Experiments were conducted on five models using three distillation attack strategies. The performance of the models was evaluated across six downstream tasks, and the ADR was used to measure the security of the models.

## Weakness
While the validity has been verified on large language models, there are currently other types of models, such as diffusion models, T5/GLM architectures, etc. Whether this framework is effective for different architectures still needs to be validated.

---

### Official Review · Reviewer_CAPj · 2025-03-01
**Review to On-Premises LLM Deployment Demands a Middle Path**

**Rating:** 8
**Confidence:** 3

**Review:**

This paper presents a principled approach to deciding which layers of black-box on-premises deployed LLMs to secure in order to balance its customizability with its protection against model distillation. The authors prove that a minimal layer exists  up to which all layers should be secured to ensure distillation robustness and propose a principled, efficient method to discover it. They confirm their experimental results with extensive experimental evaluation.

I think this paper presents a very principled approach to the proposed topic and confirms their approach with strong and convincing results. While the small targeted topic appears to be very specific and potentially not too relevant in practice yet, I think this work is a valuable contribution to making on-premises deployment a feasible option in the future.

I did not check the proof for Theorem 1 in detail, which proves that the proposed method is guaranteed to be applicable for any LLM.

I only have a few minor nitpicks:
- While very legible on colored screens, Figure 3-6 are uninterpretable on black-and-white screens/printouts.
- The "Secured Ratio" in Table 1 is never introduced, I assume from context it refers to the "number of secured layers".

---

### Official Review · Reviewer_1trD · 2025-03-02
**Review of SOLID**

**Rating:** 8
**Confidence:** 3

**Review:**

## Summary
This paper addresses the challenge of securely deploying large language models (LLMs) in privacy-sensitive environments by proposing SOLID, a semi-open framework that selectively secures a minimal set of layers to balance model confidentiality (preventing distillation attacks) and customization flexibility. The authors demonstrate that securing bottom layers of transformers provides robust security against distillation attacks while preserving fine-tuning capabilities, offering a practical solution for on-premises AI deployment in regulated industries.

## Strengths
1. Rigorous Methodology:
The paper employs a robust combination of theoretical analysis and extensive empirical validation, testing SOLID across multiple models (e.g., Llama2-70B, Mistral-7B) and distillation strategies. This thorough approach strengthens the credibility of the findings.

2. Well-Organized and Clear Explanations:
The paper is logically structured, with clear sections that guide the reader through the problem formulation, methodology, and experiments. Key concepts, such as the transition layer and distillation difficulty, are explained in an accessible manner.

3. Novel Framework and Theoretical Contribution:
SOLID is not the first to propose a partially open approach, but it advances prior work by Introducing a theoretically grounded method for selecting which layers to secure and demonstrating that securing bottom layers is more effective against distillation attacks .


## Weaknesses
1. While SOLID reduces the number of secured layers compared to fully secured models, the paper does not thoroughly discuss the computational cost of implementing hardware-secured environments (e.g., TEEs) for these layers.

---

### Decision · Program_Chairs · 2025-03-04

**Decision:**

Accept

**Comment:**

The paper is highly relevant to the workshop, as it directly addresses the challenge of trustworthy AI deployment by ensuring privacy-preserving customization while preventing model theft. The reviewers commend the paper’s rigorous theoretical foundation, extensive empirical validation across multiple models and attack strategies, and well-structured presentation.